# PM-MOE: Mixture of Experts on Private Model Parameters for Personalized Federated Learning

## Abstract

Federated learning (FL) has gained widespread attention for its privacy-preserving and collaborative learning capabilities. Due to significant statistical heterogeneity, traditional FL struggles to generalize a shared model across diverse data domains. Personalized federated learning addresses this issue by dividing the model into a globally shared part and a locally private part, with the local model correcting representation biases introduced by the global model. Nevertheless, locally converged parameters more accurately capture domain-specific knowledge, and current methods overlook the potential benefits of these parameters. To address these limitations, we propose PM-MoE architecture. This architecture integrates a mixture of personalized modules and an energy-based personalized modules denoising, enabling each client to select beneficial personalized parameters from other clients. We applied the PM-MoE architecture to nine recent model-split-based personalized federated learning algorithms, achieving performance improvements with minimal additional training. Extensive experiments on six widely adopted datasets and two heterogeneity settings validate the effectiveness of our approach. The source code is available at https://anonymous.4open.science/r/PM-MOE-8315.

## CCS Concepts

• **Computing methodologies → Distributed computing methodologies**.

## Keywords

Personalized Federated Learning; Mixture of Experts; Energy-based denoising

## 1 Introduction

The success of modern methods [9, 22, 41] is largely driven by the growing availability of training data [16, 24, 25]. Unfortunately, there are still vast amounts of isolated data remain underutilized due to strict privacy requirements [10, 42]. As a result, federated learning (FL) [1, 5, 15, 35, 39, 52], has gained significant attention for its strong privacy protection and collaborative learning capabilities. This innovative paradigm allows multiple clients to collaboratively train models, where the server only aggregates models and keep private data remaining on each client. Despite its effectiveness, traditional FL methods suffer from performance degradation due to statistical heterogeneity [20]—data domains on each client are biased, with uneven class distributions, varying sample sizes, and significant feature differences.

Personalized federated learning (PFL) [30, 31, 37, 45] alleviates this limitation by allowing each client to better fit local data. Specifically, PFL methods focus on balancing local personalization with global consistency by splitting models into global and personalized modules [40, 58], where personalized modules capture unique local

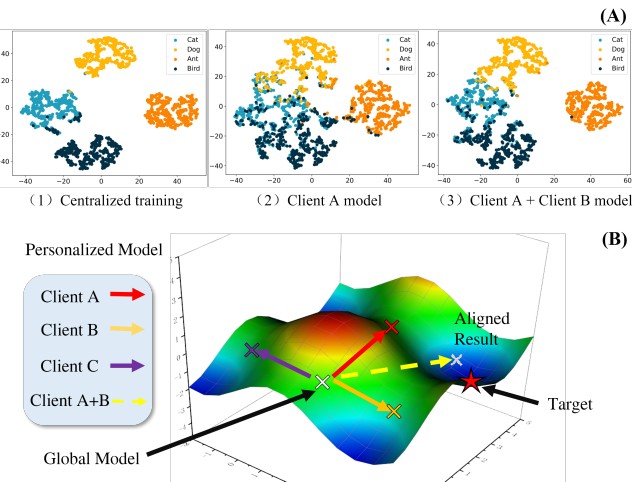

Figure 1: Motivation of our study. (A) t-SNE graph shows the inference effects of different models on the same set of data. (B) Client A gets closer to the target when using Client B's personalized model, but moves farther from the target when using Client C's personalized model.

data characteristics, mitigating global model biases and better adapting to individual client data. Recent efforts have been developed based on meta-learning [13], regularization [11, 29], model splitting [4], knowledge distillation [43–45, 49, 51], and personalized aggregation [31, 37, 57].

Given that the same types of data can be distributed across multiple clients, then a key question arises: ***"Can personalized modules from different clients mutually enhance each other's performance?"***, which is overlooked in current PFL methods. To investigate, we conducted experiments based on the state-of-the-art PFL approaches. We randomly select a client, and several data categories which distributed across different clients. Subsequently, we trained in both centralized and personalized federated learning manner. By comparison, We evaluated whether integrating personalized parameters from other clients could improve model's representation capability. As illustrated in Figure 1 (A), the selected client indeed benefited from the personalized modules from another client.

Driven by the above analysis, in this paper, we aim to explore how personalized modules from different clients can mutually enhance performance. As illustrated in Figure 1 (B), all clients utilize the same global model, while applying personalized module to debias according to the local data domain. For a single client, not all personalized modules contribute positively to the final representation. Therefore, we leverage two basic principles when designing our model: 1) Dynamically weighting the effect of personalized modules based on the current input. 2) Filtering modules that exhibit negative effects;

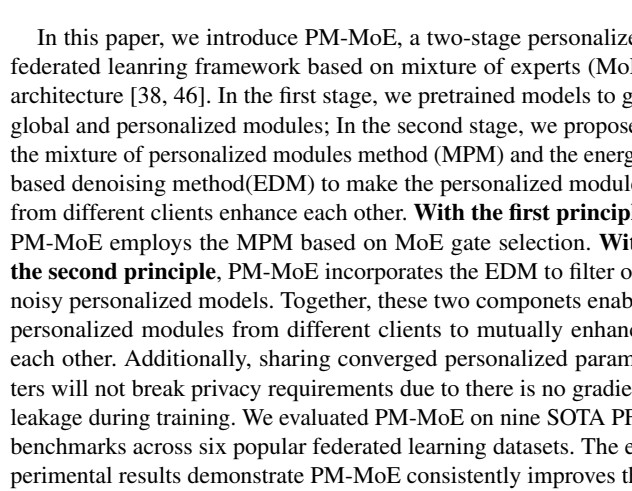

**Figure 2: Overall Architecture of Personalized Model parameters with Mixture of Experts**

In this paper, we introduce PM-MoE, a two-stage personalized federated leanring framework based on mixture of experts (MoE) architecture [38, 46]. In the first stage, we pretrained models to get global and personalized modules; In the second stage, we proposed the mixture of personalized modules method (MPM) and the energy-based denoising method(EDM) to make the personalized modules from different clients enhance each other. **With the first principle**, PM-MoE employs the MPM based on MoE gate selection. **With the second principle**, PM-MoE incorporates the EDM to filter out noisy personalized models. Together, these two componets enable personalized modules from different clients to mutually enhance each other. Additionally, sharing converged personalized parameters will not break privacy requirements due to there is no gradient leakage during training. We evaluated PM-MoE on nine SOTA PFL benchmarks across six popular federated learning datasets. The experimental results demonstrate PM-MoE consistently improves the performance of various PFL methods.

In summary, we conclude our contributions as follows:

- We propose PM-MOE, a novel two-stage framework for personalized federated learning which exchanges personalized knowledge across clients. In the first stage, the PM-MOE pretrains PFL models, followed by a fine-tuning stage for knowledge exchanges.

- Specifically, PM-MOE employs a simple MOE structure to dynamically weighting the contribution of different personalized modules. Besides, PM-MOE introduces an energy-based denosing method to filter those clients with negative effects.

- We conduct extensive experiments to nine state-of-art PFL methods across six datasets. The experimental results demonstrate PM-MOE's consistently improvement on various settings.

## 2 Notations and Preliminaries

### 2.1 Notations

In PFL, $M$ clients share the same model structure. Here, we denote notations following FedGen [47] and FedRep [17]. Each client is denoted as $C^j$ ($j \in 1, 2, ..., M$), having its own data domain $\mathcal{D}^j$ with $N^j$ samples ($j \in 1, 2, ..., M$). The data distribution of $\mathcal{D}^j$ is denoted as $P^j$. Specifically, $\mathcal{D}^j = \{x_i^j, y_i^j\}_{i=1}^{N^j}$, where $i$ is the number of training samples. $x_i$ is the $i$-th data sample and $y_i$ is its corresponding label. Each client $C^j$ in PFL has two modules: the global module and the personalized module, which is denoted as $W_g^j$ and $W_p^j$ respectively.

### 2.2 Preliminaries

In a typical PFL method, there is a centralized server who firstly aggregates clients' global modules $\left\{W_g^1, W_g^2, ..., W_g^M\right\}$, and then distributes the aggregated module $W_g$ to each client. Therefore, each client is required to firstly train on $\mathcal{D}^j$ and upload their $W_g^j$ every $E_l$ iteration. The sever aggregates global modules by the function $f$ as:

$$W_g = \frac{1}{N} \sum_{j=1}^{M} N^j f(W_g^j), \tag{1}$$

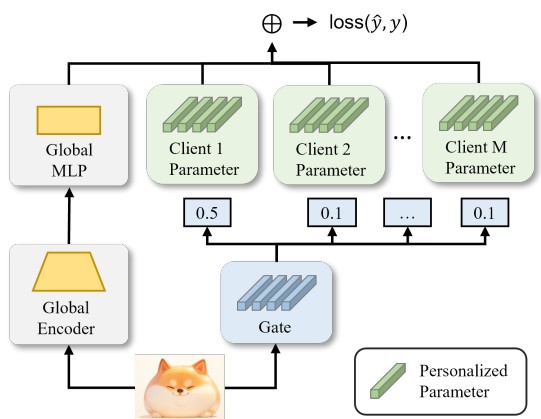

**Figure 3: Diagram of Mixture of Personalized Parameters.**

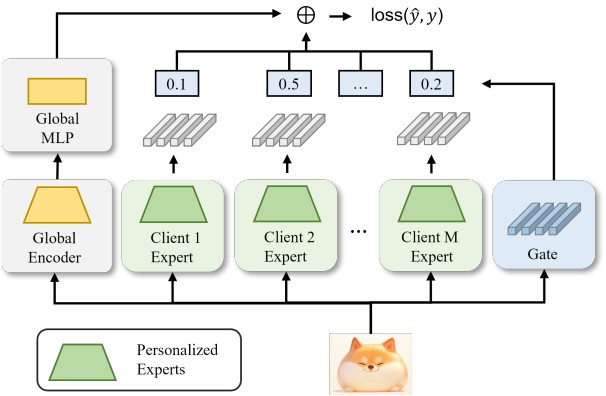

**Figure 4: Diagram of Mixture of Personalized Experts.**

where $N = \sum_{j=1}^{M} N^j$ and $f$ can be algorithms like FedAvg [39], FedProx [36], etc. After aggregation, the server sends $W_g$ to client $C^j$. Then, client $C^j$ enters the next training. Therefore, the objective loss function $\mathcal{L}$ for the entire personalized federated learning task is as follows:

$$\min_{W_g^j, W_p^j} \mathcal{L} = \min \sum_{j=1}^{M} \mathbb{E}_{(x^j, y^j) \sim P^j} [L^j(x^j, y^j; W_g^j, W_p^j)]. \quad (2)$$

Here, $L^j$ is the loss function for client $C^j$.

## 3 Method

### 3.1 The PM-MOE Overall Framework

In this section, we introduce the overall framework of PM-MOE, which which is a two-stage training framework. Specifically, our contributions lie in the mixture of personalized modules (MPM) and an energy-based denoising method (EDM). The MPM addresses the challenge of effectively utilizing personalized models, while the EDM method removes those personalized models with negative effects.

The training process of PM-MOE is divided into two steps, as shown in Figure 2. In pre-training step, we train model and obtain its converged global and personalized modules for each client, thereby constructing a personalized prompt pool. In PM-MOE step, we leverage the proposed MPM and EDM to select the optimal combination among personalized modules for each client. The following sections provide a detailed explanation of these two key phases.

*Phase 1: Pre-training.* The statistically heterogeneous distribution data $\mathcal{D}^j$ of client $C^j$ is mapped to the feature space $x_{g,rep}^j$ through the global feature extractor $f_g : \mathbb{R}^U \rightarrow \mathbb{R}^D$, and to the feature space $x_{p,rep}^j$ via the personalized feature extractor $f_p : \mathbb{R}^U \rightarrow \mathbb{R}^D$. The weighted aggregated feature space $x_{rep}^j = x_{g,rep}^j + x_{p,rep}^j$ is then mapped to the corresponding label space through the global classifier $s_g : \mathbb{R}^D \rightarrow \mathbb{R}^C$ and the personalized classifier $s_p : \mathbb{R}^D \rightarrow \mathbb{R}^C$. $U$, $D$ and $C$ represent the input space, feature space, and label space, respectively.

$$x_{rep}^j = f_g \left( W_{g,fe}^j, x^j \right) + f_p \left( W_{p,fe}^j, x^j \right). \quad (3)$$

Additionally, as seen in DBE [55], there exists a personalized vector parameter $PP^j \in \mathbb{R}^D$ to correct the local data distribution. The associated expressions are as follows:

$$\hat{y}^j = s_g \left( W_{g,hd}^j, h^j \right) + s_p \left( W_{p,hd}^j, h^j \right) + PP^j. \quad (4)$$

During training, the global model parameters $W_{g,fe}^j$ and $W_{g,hd}^j$ are uploaded to the server for aggregation, while the personalized model parameters $W_{p,fe}^j, W_{p,hd}^j$ and $PP^j$ are computed locally and not uploaded. After the global training process with $E_g$ epochs, the model converges.

*Phase 2: PM-MOE Fine-Tuning.* First, after the convergence of the model-splitting-based series of models, the server collects the trained personalized model parameters to form a personalized parameter pool, which is then distributed to each client. Next, each client locally trains a gating network, which assigns weights to each personalized model based on the input data, thereby effectively utilizing the personalized knowledge from all clients. For detailed information, refer to Section 3.2. Finally, since some of the personalized knowledge from other clients is irrelevant to the local data distribution, training the gating network with these noisy parameters can hinder convergence. To address this, we designed an energy-based denoising method. For further details, see Section 3.3.

### 3.2 Mixture of Personalized Modules

PM-MOE is a flexible architecture, and to accommodate the complex and diverse model-splitting-based personalized federated learning algorithms, we designed two adaptation methods: MPP and MPE, as shown in Figures 3 and 4.

Assume that a personalized federated learning algorithm involves personalized parameters, these parameters do not project data into vectors of other dimensions. We define this type as local personalized parameters ($PP$). Suppose the personalized federated learning algorithm also involves personalized expert models, where the expert models $\mathcal{W}_p^j$ map data $\mathcal{D}^j$ to a new feature space. We define this type as local personalized experts ($PE$). The server builds and synchronizes a set of personalized models. Depending on the type of personalized model, the server collects the converged model parameters from all clients, constructing a personalized parameter pool

$\mathcal{W}_{PP} = \{PP^j\}_{j=1}^M$ and a personalized expert pool $\mathcal{W}_{PE} = \{W_{PE}^j\}_{j=1}^M$. The server then synchronizes these sets with all clients.

Clients build a gating network and fine-tune parameters. Since each personalized federated learning client is diverse, as shown in Figure 3, we divide the combination of the gating network and personalized models into two categories. **The first is commonalities.** The calculation of set weights depends on the input data $x^j$. To achieve this, we construct gating networks $G_{PP}^j, G_{PE}^j$ for the personalized parameter and the personalized expert, with corresponding training parameters $\theta_{PP}^j, \theta_{PE}^j$. The weight calculations are formally represented as follows:

$$\alpha_{PP} = G_{PP}^j\left(x^j, \theta_{PP}^j\right) \tag{5}$$

$$\alpha_{PE} = G_{PE}^j\left(x^j, \theta_{PE}^j\right) \tag{6}$$

We then sort the weights calculated by formulas (1) and (2) in descending order. From the set, we select the top k parameters and construct the personalized parameter and expert subsets as $\{\alpha_{PP}^l\} = Top\,(k, \alpha_{PP}), \{\alpha_{PE}^l\} = Top\,(k, \alpha_{PE})$. Here, $l$ denotes the index of the selected clients, where $l \in [1, M]$.

**Second is Differences.** Since the personalized parameter pool $\mathcal{W}_{PP}$ does not process data, we directly compute the weighted sum of the personalized parameters, resulting in a vector with the same shape as the local personalized parameter $PP^j$ as follows:

$$PP_{moe}^j = \mathcal{W}_{PP}^l \cdot \alpha_{PP}^l \tag{7}$$

In our setting, the weighted vector $\{PP_{moe}\}^j$ replaces the local personalized parameter $\{PP\}^j$ on client $C^j$. For the personalized expert set $\mathcal{W}_{PE} = W_{PE\,j=1}^{j\,M}$, taking the personalized classifier $s_p$ as an example, each expert maps the data to a new feature space $h^l$, where:

$$h^l \in \mathbb{R}^C = s_p^l(x^j, W_{PE}^l) \tag{8}$$

We then compute the mixed weighted personalized parameter vector $x_{moe}^j$ as:

$$x_{moe}^j = h^l \cdot \alpha_{PE}^l \tag{9}$$

The client $C^j$ then replaces the output of the local personalized expert $W_{PE}^j$ with $x_{moe}^j \in \mathbb{R}^C$.

Since the converged parameters reflect the local data knowledge that each client has spent significant effort training, during the training process, the personalized parameter and expert sets $\mathcal{W}_{PP}, \mathcal{W}_{PE}$ are frozen and not optimized together with the gating network.

### 3.3 Energy-based Personalized Modules Denoising

Due to MOE using Top-K to select appropriate experts, this ranking based solely on parameter magnitude lacks confidence and introduces noise to some extent. It leads to the gating network optimizing gradients in the wrong direction. To effectively remove noise from the personalized parameter pool, inspired by energy-based models, we propose an energy-based personalized expert denoising method.

The core idea is to build an energy function to describe the dependency or similarity between inputs. Simply put, the essence of the method is to calculate energy—high energy corresponds to low similarity, while low energy indicates high similarity. Taking the personalized feature extractor experts as an example, for client $C^j$, the personalized expert pool $\mathcal{W}_{PE} = \{W_{PE}^j\}_{j=1}^M$ uses a projection

function $f_p : \mathbb{R}^U \to \mathbb{R}^D$ to map data $x^j$ to $H = \{h^1, h^2, ...h^j, ...h^M\}$. The vector $h^j \in \mathbb{R}^D$ from the local client is taken as the scalar for energy. Then, the vectors mapped by other client models are projected into the coordinate system of $h^j$. Let's define the energy function for a given input pair $(h^j, h^k)$ as follows:

$$E^k(h^j, h^k) = -v^k\,[\mathbb{I}] \tag{10}$$

where $v^k \in \mathbb{R}^D = \frac{h^k}{h^j}$, $(k \neq j)$. And each dimension of the vector is denoted by $\mathbb{I} \in D$. The projected vector set is defined as $V = \{v^1, ..., v^k\}, (k \in [M], k \neq j)$. Then, Helmholtz free energy can be expressed as the negative logarithm of the partition function:

$$F_T^k(v^k) = -T log \sum exp(-E^k(h^j, h^k)/T) \tag{11}$$

Since we use the local client's vector as a fixed anchor, it is natural to choose the negative Helmholtz free energy as the confidence score for the similarity between $h^k$ and $h^j$

$$H^k(h^j, h^k) = -F_T^k(v^k) = T log \sum exp(-E^k(h^j, h^k)/T) \tag{12}$$

where $T$ is the temperature parameter. Therefore, we use the confidence score to filter out noise (irrelevant experts). Then, we set a dropout ratio coefficient $\gamma \in (0, 1)$. The confidence scores of all personalized feature extraction experts are sorted in ascending order, and the bottom $\gamma$-proportion of experts are removed.

Finally, after the two modules of PM-MOE framework, the total objective loss of PM-MOE is as follows:

$$\min_{\theta_{PP}^j, \theta_{PE}^j} \mathcal{L} = min \sum_j^K \mathbb{E}_{(x^j, y^j)\,Pj} L^j(x^j, y^j; \theta_{PP}^j, \theta_{PE}^j) \tag{13}$$

### 3.4 Theoretical Analysis

In this section, we demonstrate that **leveraging personalized models converged from other clients is more beneficial for improving the performance of local models.** In simple terms, model-split-based personalized federated learning shares uploaded models to learn from the data distribution of all parties involved. However, the heterogeneous nature of the data creates a tug-of-war, resulting in inefficient knowledge transfer between clients. Interestingly, the private parameters that are not uploaded by clients best capture local knowledge.

Therefore, we propose that utilizing these converged personalized models is necessary to enhance performance, leading to the design of the PM-MOE architecture. In this subsection, we theoretically prove that the PM-MOE architecture converges to a lower bound. Even in extreme cases, where each client's data distribution is entirely different, this architecture does not degrade the performance of local models.

THEOREM 3.1. (*Lower Bound on the Final Accuracy of MPE*) *Suppose there are $M(\geq 2)$ client experts predicting independently, each with an average accuracy rate of $p(> 0)$. If a trained gate network assigns samples to the client experts such that the ratio of the probability of assigning a sample to a correct expert versus an incorrect expert is $1 + \alpha$, where $\alpha > 0$. Then, the final accuracy of MPE is bounded from below by:*

$$P_{MPE} \geq \frac{(1+\alpha)p}{1 + \alpha(p + \frac{1-p}{M})} > p = P_{client}. \tag{14}$$

**Table 1: Results of federated and personalized federated learning algorithms on six datasets with heterogeneous data distribution (Dirichlet distribution with $S = 0$ and $S = 20$). Bold: Best performance.**

| Spilt Type | $S = 0$ | | | | | | $S = 20$ | | | | | |
|---|---|---|---|---|---|---|---|---|---|---|---|---|
| Method | MNIST | FMNIST | Cifar10 | Cifar100 | TINY | AGNews | MNIST | FMNIST | Cifar10 | Cifar100 | TINY | AGNews |
| FedAvg | 98.93 | 88.64 | 63.68 | 32.94 | 17.69 | 62.40 | 98.95 | 90.69 | 67.74 | 35.37 | 19.66 | 71.68 |
| FedProx | 98.93 | 88.50 | 63.85 | 33.07 | 17.60 | 65.75 | 98.98 | 90.79 | 67.54 | 35.42 | 19.56 | 72.90 |
| SCAFFOLD | 99.12 | 88.74 | 64.19 | 34.71 | 19.67 | 78.85 | 99.18 | 91.44 | 70.40 | 38.54 | 19.67 | 78.50 |
| FedGEN | 98.98 | 88.79 | 64.36 | 32.72 | 15.85 | 63.13 | 98.98 | 90.90 | 67.59 | 34.52 | 17.70 | 71.65 |
| MOON | 98.92 | 88.59 | 63.87 | 33.00 | 17.57 | 62.21 | 98.98 | 90.74 | 67.53 | 35.36 | 17.57 | 71.76 |
| FedPer | 99.49 | 97.53 | 89.90 | 48.27 | 36.36 | 93.99 | 98.62 | 93.57 | 76.67 | 36.28 | 25.91 | 88.75 |
| LG-FedAvg | 99.28 | 97.25 | 89.02 | 47.03 | 33.20 | 94.33 | 97.85 | 92.23 | 73.95 | 35.90 | 23.78 | 87.77 |
| FedRep | 99.46 | 97.58 | 90.19 | 49.44 | 38.09 | 93.78 | 98.61 | 93.77 | 77.25 | 36.52 | 25.77 | 89.02 |
| FedRoD | 99.68 | 97.60 | 90.07 | 51.92 | 38.90 | 93.65 | 99.33 | 94.06 | 79.50 | 42.45 | 29.07 | 88.91 |
| FedGH | 99.29 | 97.40 | 84.50 | 48.61 | 25.80 | 92.58 | 97.94 | 92.25 | 73.79 | 37.88 | 21.51 | 88.19 |
| FedBABU | 99.67 | 97.74 | 91.38 | 50.83 | 34.53 | 92.87 | 99.32 | 94.71 | 82.17 | 40.46 | 25.92 | 87.58 |
| GPFL | 99.49 | 94.91 | 77.79 | 57.41 | 27.08 | 90.84 | 99.49 | 93.21 | 72.39 | 49.01 | 22.92 | 82.41 |
| FedCP | 99.75 | 98.31 | 93.76 | 69.83 | 65.97 | 92.40 | 99.27 | 94.45 | 80.29 | 41.79 | 31.93 | 87.62 |
| DBE | 98.17 | 93.95 | 89.11 | 60.33 | 38.29 | 93.70 | 96.97 | 91.11 | 79.76 | 52.30 | 31.11 | 89.08 |
| **PM-MOE** | **99.85** | **98.61** | **93.95** | **70.68** | **66.33** | **94.76** | **99.49** | **94.79** | **82.21** | **52.36** | **32.15** | **89.16** |

We briefly proof the key steps, and other details are given in the appendix. □

PROOF. We define the event set:

$\mathcal{A} := \{s$ out of $M$ experts are able to predict correctly$\}$.

$$P(\mathcal{A}) = \binom{M}{s} p^s (1-p)^{M-s}, \qquad (15)$$

Under the above condition, if the gated network assigns s client weights, the model is able to still correctly predict the sample. We have:

$$P(\mathcal{B} \mid \mathcal{A}) = \frac{(1+\alpha)s}{(1+\alpha)s + (M-s)} = \frac{(1+\alpha)s}{M + \alpha s}, \qquad (16)$$

And then, $\mathcal{B} := \{$MPE can predict correctly$\}$. We have:

$$P(\mathcal{B}) = \sum_{\mathcal{A}} P(\mathcal{A})P(\mathcal{B} \mid \mathcal{A})$$
$$= \mathbb{E}\left[ \frac{(1+\alpha)Mp}{M + \alpha(t+1)} \right] \quad (\text{Let } t = s - 1). \qquad (17)$$

Define the function:

$$f(t) = \frac{(1+\alpha)Mp}{M + \alpha(t+1)}. \qquad (18)$$

We observe that $f(t)$ is a convex function. And by Jensen's inequality, we have:

$$\mathbb{E}[f(t)] \geq f(\mathbb{E}[t]). \qquad (19)$$

Combining (17) and (19) yields

$$P_{\text{MPE}} = P(\mathcal{B}) \geq f(\mathbb{E}[t]) = \frac{(1+\alpha)p}{1 + \alpha(p + \frac{1-p}{M})}. \qquad (20)$$

The last expression is strictly increasing with respect to $\alpha$ when $\alpha > 0$, and thus:

$$P_{\text{MPE}} \geq \frac{(1+\alpha)p}{1 + \alpha(p + \frac{1-p}{M})} > \frac{(1+0)p}{1 + 0(p + \frac{1-p}{M})} = p = P_{\text{client}}. \qquad (21)$$

Theorem 3.1 indicates that the accuracy of the MPE is bounded below by the average accuracy of an individual client. Furthermore, the lower bound $\frac{(1+\alpha)p}{1+\alpha(p+\frac{1-p}{M})}$ increases monotonically with respect to both $\alpha$ and $M$. In other words, the accuracy of the MPE will be improved as the training of the gate network. Specifically, when the gate network is well trained (i.e., $\alpha \gg 0$) and $M$ is large enough, the accuracy of MPE will asymptotically approach 100%.

## 3.5 Privacy Analysis

For model-splitting-based personalized federated learning algorithms combined with PM-MOE, data privacy is ensured in both phases.

**In the pre-training phase**, each client uploads only the shared parameters to the server, while personalized parameters are trained locally. Due to the model splitting, the link between shared and personalized parameters is severed. The gradient information of personalized parameters remains private to each client, making it difficult to breach data privacy through model inversion attacks [3].

**In the PM-MOE phase**, both the server and clients only receive the converged personalized model parameters. Clients cannot infer the training data or other private information from the model parameters. Therefore, the proposed approach effectively safeguards data privacy.

## 4 Experiment

### 4.1 Experiment Setup

In this section, we compare the proposed PM-MOE with nine state-of-the-art model-splitting-based personalized federated learning algorithms. We validate the effectiveness of our method across six datasets. Additionally, we conduct comprehensive ablation studies on these algorithms to better determine parameter choices and the impact of various components.

**Table 2: Ablation experiments of PM-MOE across 9 state-of-the-art model-split-based personalized federated learning algorithms.**

| Spilt Type | S = 0 | | | | | | S = 20 | | | | | | Avg ↑ |
|---|---|---|---|---|---|---|---|---|---|---|---|---|---|
| Method | MNIST | FMNIST | Cifar10 | Cifar100 | TINY | AGNews | MNIST | FMNIST | Cifar10 | Cifar100 | TINY | AGNews | |
| FedPer | 99.49 | 97.53 | 89.90 | 48.27 | 36.36 | 93.99 | 98.62 | 93.57 | 76.67 | 36.28 | 25.91 | 88.75 | - |
| +PM-MOE | **99.50** | **97.55** | **89.96** | **48.34** | **36.42** | **93.99** | **98.62** | **93.59** | **76.69** | **36.30** | **25.96** | **88.75** | 0.0275 |
| LG-FedAvg | 99.28 | 97.25 | 89.02 | 47.03 | 33.20 | 94.33 | 97.85 | 92.23 | 73.95 | 35.90 | 23.78 | 87.77 | - |
| +PM-MOE | **99.28** | **97.28** | **89.19** | **47.10** | **33.25** | **94.76** | **97.85** | **92.23** | **74.01** | **35.90** | **23.78** | **87.77** | 0.0325 |
| FedRep | 99.46 | 97.58 | 90.19 | 49.44 | 38.09 | 93.78 | 98.61 | 93.77 | 77.25 | 36.52 | 25.77 | 89.02 | - |
| +PM-MOE | **99.47** | **97.60** | **90.24** | **49.49** | **38.12** | **93.87** | **98.61** | **93.82** | **77.25** | **36.52** | **25.78** | **89.02** | 0.0258 |
| FedRoD | 99.68 | 97.60 | 90.07 | 51.92 | 38.90 | 93.65 | 99.33 | 94.06 | 79.50 | 42.45 | 29.07 | 88.91 | - |
| +PM-MOE | **99.69** | **97.66** | **90.22** | **52.82** | **39.25** | **93.72** | **99.33** | **94.06** | **79.50** | **42.60** | **29.07** | **88.93** | 0.1425 |
| FedGH | 99.29 | 97.40 | 84.50 | 48.61 | 25.80 | 92.58 | 97.94 | 92.25 | 73.79 | 37.88 | 21.49 | 88.10 | - |
| +PM-MOE | **99.30** | **97.40** | **88.61** | **48.69** | **25.82** | **92.66** | **97.94** | **92.25** | **73.79** | **37.88** | **21.51** | **88.19** | 0.3675 |
| FedBABU | 99.67 | 97.74 | 91.38 | 50.83 | 34.53 | 92.87 | 99.32 | 94.71 | 82.17 | 40.46 | 25.92 | 87.58 | - |
| +PM-MOE | **99.67** | **97.76** | **91.41** | **50.83** | **34.53** | **93.55** | **99.32** | **94.79** | **82.17** | **40.46** | **25.92** | **88.29** | 0.1267 |
| GPFL | 99.49 | 94.91 | 77.79 | 57.41 | 27.08 | 90.84 | 99.49 | 93.21 | 72.39 | 49.01 | 22.92 | 82.41 | - |
| +PM-MOE | **99.50** | **95.56** | **82.28** | **57.41** | **27.08** | **91.94** | **99.49** | **93.21** | **72.39** | **49.01** | **22.92** | **82.41** | 0.5208 |
| FedCP | 99.75 | 98.31 | 93.76 | 69.83 | 65.97 | 92.40 | 99.27 | 94.45 | 80.29 | 41.79 | 31.93 | 87.62 | - |
| +PM-MOE | **99.85** | **98.61** | **93.95** | **70.68** | **66.33** | **92.48** | **99.30** | **94.63** | **80.51** | **42.49** | **31.96** | **87.67** | 0.2575 |
| DBE | 98.17 | 93.95 | 89.11 | 60.33 | 38.29 | 93.70 | 96.97 | 91.11 | 79.76 | 52.30 | 31.10 | 89.08 | - |
| +PM-MOE | **99.63** | **97.23** | **89.90** | **60.53** | **38.36** | **93.74** | **99.38** | **93.40** | **80.05** | **52.30** | **31.11** | **89.08** | 0.9033 |

*Dataset and Data Partitioning.* To effectively compare the performance of the proposed approach, we use six widely-adopted benchmark datasets for personalized federated learning: AG News [60], Cifar10 [23], Cifar100 [23], MNIST [26], Fashion-MNIST [50], and Tiny-ImageNet [7]. Specifically, we adopt a Dirichlet distribution [33] with a shared ratio $S(0 < S < 100)$ for data partitioning.

- **Dirichlet distribution with** $S = 20$: In the first setting, 20% of the data for each class is uniformly distributed among $M$ clients, and the remaining data is assigned based on Dirichlet-distributed weights.

- **Dirichlet distribution with** $S = 0$: In the second setting, no constraints are placed on class distribution across clients, with all data allocated based on Dirichlet-distributed weights.

*Baseline Methods.* We referred to the Personal Federated Learning library, PFLlib [59]. Furthermore, we compared general federated learning algorithms such as FedAvg [39], FedProx [36], SCAFFOLD [21], MOON [28], and FedGen [62], alongside recent state-of-the-art personalized federated learning methods, including personalized feature extractors like FedGH [53], LG-FedAvg [32], FedBABU [40], FedCP[58], GPFL [56], FedPer[4], FedRep [8], FedRod [6], and DBE [55] for personalized parameters.

*Experimental Details.* To ensure fairness, we employ a 4-layer CNN model as the backbone for Cifar10, Cifar100, MNIST [26], Fashion-MNIST [50], and Tiny-ImageNet [7] datasets, and a fast-Text [19] model for AG News. Each personalized model is pre-trained for 2000 epochs until convergence. We optimize three key parameters: $\eta_{moe}$ (local MOE learning rate), $k$ (top k MOE weights), and $E_{moe}$ (local MOE training iterations). All experiments are executed on a single RTX 3090 GPU.

## 4.2 Experimental Results

*Main Results.* Tables 1 show that PM-MOE consistently outperforms other personalized federated learning methods across both partitioning settings in tasks ranging from 4 to 200 classes. Compared to traditional federated learning methods, personalized federated learning better handles data heterogeneity, with a performance improvement of up to 48.64% over the FedAvg baseline. Interestingly, when data heterogeneity decreases, the overall performance of personalized methods also drops. The PM-MOE framework leverages the personalized models converged from all clients to improve each client's performance. If parameters from other clients are noisy, the local gating network assigns weights to prioritize the local personalized model, protecting its performance. Conversely, if external parameters are useful, the network allocates weights accordingly, enhancing the local model's performance.

*Analysis of Gating Network Parameters.* As the most critical component for each client in this framework is the training of the gating network, this section presents parameter experiments focused on tuning the number of layers, activation functions, and initialization parameters of the gating network. In this subsection, to highlight the differences between methods across different dimensions, we will apply sigmoid normalization to the data in the experimental group.

- **Number of Layers in the Gating Network**: We conducted four sets of experiments on the number of layers in the gating network. 1 layer: (input dimension, number of experts); 2 layers: (input dimension, 128, number of experts); 3 layers: (input dimension, 128, 256, number of experts); 4 layers: (input dimension, 128, 256, 128, number of experts). As shown in Figure 5-(a), increasing the depth of the gating network proves to be effective. The gating

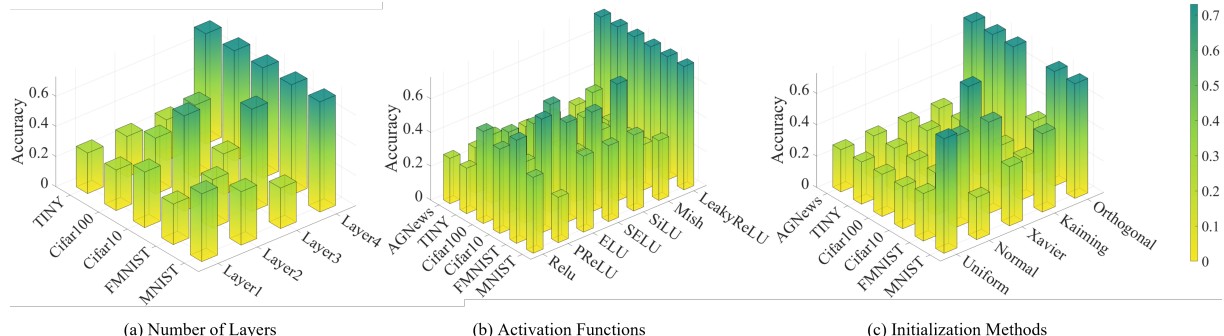

**Figure 5: Results of Gating Network Parameters**

**Table 3: Ablation Experiment Analysis Results**

| Method | AGNews | FMNIST | Cifar100 | Avg ↑ |
|---|---|---|---|---|
| pFL | 94.33 | 98.31 | 69.83 | - |
| +MOE | 94.52 | 98.39 | 70.12 | 0.19% |
| +MOE+Denoising | 94.76 | 98.61 | 70.68 | 0.53% |

network needs to determine the weights for all personalized parameters based on input data, requiring deeper neural networks on the client side to capture data features effective.

- **Activation Functions of the Gating Network**: For the 4-layer feedforward gating network, we tested common neural network activation functions such as ReLU, LeakyReLU, PReLU, ELU, SELU, SiLU, and Mish. As shown in Figure 5-(b), the most effective activation function is LeakyReLU. LeakyReLU's non-linearity in the negative region allows the neural network to learn and model more complex data, effectively assigning personalized model parameters the appropriate weights.

- **Initialization Methods for the Gating Network**: We compared commonly used parameter initialization methods, including uniform distribution, normal distribution, Xavier, Kaiming, Orthogonal, and Spectral. As shown in Figure 5-(c), most results indicated that the Orthogonal initialization method yields the best performance for gated network models. This method draws initial weights from the orthogonal group, maintaining dynamic isometry throughout the network's learning process, which helps preserve a relatively stable proportional relationship between input and output signals.

*Ablation Study.* In this section, we conducted ablation studies to evaluate the effectiveness of each individually designed module. The experiments confirmed that the PM-MOE component and the denoising component improve the performance of the model-split-based personalized federated learning algorithm. We selected the best performing personalized federated learning methods in the comparison dataset for comparison. As shown in Table 3, adding the MOE component resulted in an average improvement of 0.2% across the 4, 10, and 100 class settings. The regular denoising ratio was set to 0.2, and adding the denoising component led to an average improvement of 0.53%.

In detail, we demonstrated that our proposed PM-MOE framework improves nine state-of-the-art personalized federated learning algorithms. Specifically, we used data heterogeneity settings of $S = 0$ (100% heterogeneity) and $S = 20$ (80% heterogeneity). As shown in Table 2, PM-MOE enhances the performance of all personalized federated learning algorithms across six widely adopted datasets.

*Model Parameter Analysis.*

- **Top-k Impact Analysis**: The number of personalized parameters determines the breadth of knowledge. If top $k$ is too small, it may not fully utilize knowledge from other clients. If top $k$ is too large, it may introduce excessive noisy knowledge. Therefore, we conducted extensive experiments on the choice of $k$. Keeping other conditions constant, we set $k = 2, 4, 8, 16, 20$ across 20 clients. As shown in Figure 6, in highly heterogeneous data settings ($S = 0$), many clients do not share categories. Thus, setting $k$ to half the number of clients helps the gating network select more effective personalized parameters. In settings with some shared data ($S = 20$), where each client shares a few categories, a larger $k$, typically equal to the number of clients, is preferable as it allows the gating network to reference personalized knowledge from all clients.

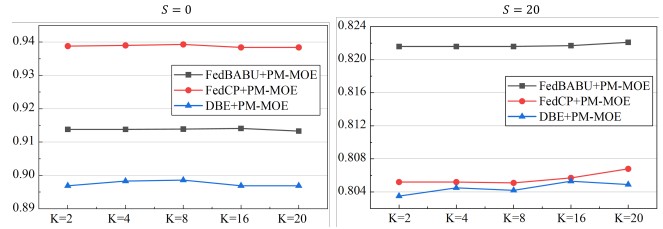

**Figure 6: Impact of Top k in PM-MOE.**

- **Gating Network Learning Rate Analysis**: Keeping other variables constant, we set the learning rate of the gating network $\eta_{moe}$ to 0.05, 0.1, and 0.5. As shown in Figure 7, for both $S = 0$ and $S = 20$ heterogeneous data settings, the MOE gating network should be assigned a higher learning rate. A smaller learning rate may cause the model to get stuck in local minima or saddle points, leading to worse performance.

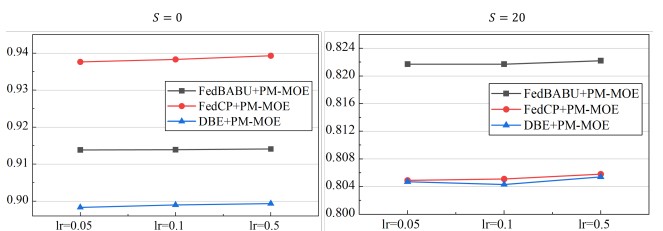

**Figure 7: Impact of Gating Network Learning Rate in PM-MOE.**

- **Impact of MOE Training Iterations**: We further explored whether the number of local training iterations affects PM-MOE. As shown in Figure 8, after adding PM-MOE to three algorithms, performance slightly decreases with more training iterations, possibly due to overfitting. Therefore, in the case of converged pre-trained personalized federated learning, training for 50 epochs per client is sufficient.

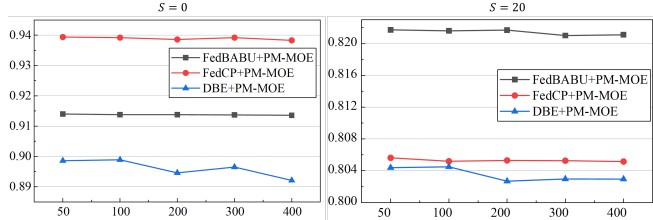

**Figure 8: Impact of Training Epochs in PM-MOE.**

## 4.3 Analysis of the combination of personalized federated learning and MOE

For integrating MoE in personalized federated learning, both PFL-MoE and FedMoE use gated networks to adjust the weight balance between local personalized models and the global model. Both methods train the local models synchronously. For fair comparison, we employed a gated network to balance the global and local models' weights, training it synchronously with gradient optimization, referred to as synchronous MoE. The pre-training phase lasted 2000 rounds, with results presented in Table 4. While performance degradation in synchronous MoE is minor for the 4-class AGNews dataset, it becomes significant as task complexity increases, particularly for datasets with 10, 100, or 200 classes like FMNIST, CIFAR-10, CIFAR-100, and TINY. This decline likely occurs because synchronous training forces the gated network to balance unconverged global and local parameters, making it more susceptible to noise and assigning suboptimal weights, which degrades overall model performance.

## 5 Related Work

*Personalized Federated Learning.* Personalized Federated Learning (PFL) was introduced to address the limitations of traditional federated learning in handling non-IID data and personalized requirements. PFL employs various strategies such as regularization [11, 29], meta-learning [13], knowledge distillation [43–45, 49, 51], model splitting [4], and personalized aggregation [31, 37, 57].

**Table 4: Moe Combination Analysis Results**

| Method | MNIST | FMNIST | Cifar10 | Cifar100 | TINY | AGNews |
|---|---|---|---|---|---|---|
| Fed-Syn-MoE | 89.48 | 91.47 | 78.31 | 18.75 | 13.59 | 93.37 |
| PM-MOE | 99.85 | 98.61 | 93.95 | 70.68 | 66.33 | 94.76 |

pFedMe [11] leverages the convexity and smoothness of Moreau Envelopes to facilitate its convergence analysis, while Per-FedAvg [13] incorporates meta-learning into federated learning. FedDistill [43] transfers global knowledge to local models through distillation.

*Model-Splitting-Based Personalized Federated Learning.* Model-splitting-based personalized federated learning has recently gained traction by balancing personalization and global consistency through model partitioning. These methods fall into three categories: the first combines personalized feature extractors with a globally shared classifier, as seen in FedGH[53] and LG-FedAvg [4], allowing clients to maintain unique feature extraction while ensuring consistency through a shared classifier. The second type uses a globally shared feature extractor and personalized classifiers, as demonstrated by FedBABU [40], FedCP [58], GPFL [56], FedPer [4], FedRep [8], and FedRod [6], enabling client-specific adaptation while preserving shared feature extraction. The third type, such as DBE

*Personalized Federated Learning and MOE.* In personalized federated learning, methods integrating Mixture of Experts [18, 61] (MoE) models, such as PFL-MoE [14] and FedMoE [54], deploy both local feature extractors (local experts) and globally shared feature extractors (global experts) on each client, with a gating network controlling the output weights of these experts. PFL-MoE primarily addresses homogeneous models, modulating the experts' weights via the gating network. In contrast, FedMoE emphasizes model heterogeneity by incorporating experts with more parameters than the global model to better capture local data characteristics.

*Energy-based denoising methods.* Energy-based models [27] (EBMs) capture variable dependencies by assigning scalar energy values to each input configuration. EBMs have been applied across various domains, including generative modeling [12], out-of-distribution detection [34], open-set classification [2], and incremental learning [48]. In personalized federated learning, EBMs are able to quantify relationships between model parameters using energy as a metric. For selecting personalized experts in Mixture of Experts (MoE) models, EBMs can filter out ineffective experts, thus denoising the model. Despite its potential, the use of EBMs for expert denoising in MoE remains underexplored.

## 6 Conclusion

In this article, we propose the PM-MOE framework to integrate the construction of a personalized parameter pool with local MOE training. PM-MOE aggregates the converged private model parameters from all clients, allowing each client to selectively reference the knowledge of others. This architecture effectively enhances the ability of model-splitting-based personalized federated learning algorithms to learn global knowledge. Through extensive experiments and theoretical analysis, we demonstrate the superiority of PM-MOE.

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

# A  Appendix

## A.1  Theoretical Derivations

PROOF. Considering $M$ client experts predict independently, the probability that exactly $s$ experts can predict correctly is given by the binomial distribution $\text{Bin}(M, p)$:

$$P(\mathcal{A}) = \binom{M}{s} p^s (1-p)^{M-s}, \tag{22}$$

where the event set $\mathcal{A} := \{s \text{ out of } M \text{ experts can predict correctly}\}$. Under the above condition, if the gate network can assign a sample to any of these $s$ client experts, then the MPE can predict the sample correctly. Therefore, we have:

$$P(\mathcal{B} \mid \mathcal{A}) = \frac{(1+\alpha)s}{(1+\alpha)s + (M-s)} = \frac{(1+\alpha)s}{M + \alpha s}, \tag{23}$$

where $\mathcal{B} := \{\text{MPE can predict correctly}\}$. According to the law of total probability, the probability that the MPE can predict correctly is:

$$
\begin{aligned}
P(\mathcal{B}) &= \sum_{\mathcal{A}} P(\mathcal{A}) P(\mathcal{B} \mid \mathcal{A}) \\
&= \sum_{s=0}^{M} \binom{M}{s} p^s (1-p)^{M-s} \frac{(1+\alpha)s}{M + \alpha s} \\
&= \sum_{s=1}^{M} \binom{M-1}{s-1} p^{s-1}(1-p)^{M-s} \frac{(1+\alpha)Mp}{M + \alpha s} \\
&= \sum_{t=0}^{M-1} \binom{M-1}{t} p^t (1-p)^{M-1-t} \frac{(1+\alpha)Mp}{M + \alpha(t+1)} \quad (\text{Let } t = s-1) \\
&= \mathbb{E}\left[\frac{(1+\alpha)Mp}{M + \alpha(t+1)}\right].
\end{aligned}
\tag{24}
$$

Here, the expectation is taken over $t$ which follows $\text{Bin}(M-1, p)$. Define the function:

$$f(t) = \frac{(1+\alpha)Mp}{M + \alpha(t+1)}. \tag{25}$$

We observe that $f(t)$ is a convex function with respect to $t$ because its second derivative is positive:

$$f''(t) = \frac{2(1+\alpha)Mp\alpha^2}{(M + \alpha t + \alpha)^3} > 0, \tag{26}$$

since $\alpha, p, M > 0$. By Jensen's inequality, we have:

$$\mathbb{E}[f(t)] \geq f(\mathbb{E}[t]). \tag{27}$$

The expected value of $t$ is:

$$\mathbb{E}[t] = (M-1)p. \tag{28}$$

Combining (24), (27) and (28) yields

$$
\begin{aligned}
P_{\text{MPE}} = P(\mathcal{B}) \geq f(\mathbb{E}[t]) &= f((M-1)p) \\
&= \frac{(1+\alpha)Mp}{M + \alpha((M-1)p+1)} \\
&= \frac{(1+\alpha)p}{1 + \alpha(p + \frac{1-p}{M})}.
\end{aligned}
\tag{29}
$$

The last expression is strictly increasing with respect to $\alpha$ when $\alpha > 0$, and thus:

$$P_{\text{MPE}} \geq \frac{(1+\alpha)p}{1 + \alpha(p + \frac{1-p}{M})} > \frac{(1+0)p}{1 + 0(p + \frac{1-p}{M})} = p = P_{\text{client}}. \tag{30}$$

$\square$

## A.2  Algorithm

Following the design principles of MPP, MPE and denoising module, we apply this algorithm to nine state-of-the-art model-splitting-based personalized federated learning algorithms. The general training process is outlined in Algorithm 1.

---
**Algorithm 1** Personalized Model Training with MOE

---
**Input:** $M$: Number of clients; $W_{g,fe}$: Pre-trained parameters of the global feature extractor; $W_{g,hd}$: Pre-trained parameters of the global head; $\{W_{p,fe}^j, W_{p,hd}^j\}_{j=1}^{M}$: Pre-trained parameters of the client $j$ personalized head; $\eta_{moe}$: Local MOE learning rate; $k$: Top-k value for MOE weights; $E_{moe}$: Local MOE training iterations; $\theta_G^j$: Client $j$'s model parameters of the gating network; $\mathcal{W}_{PE}, \mathcal{W}_{PP}$: Personalized expert set and personalized parameter set, respectively. $\gamma$: Dropout ratio.

**Output:** $\{\theta_{PE}^1, \ldots, \theta_{PE}^M\}, \{\theta_{PP}^1, \ldots, \theta_{PP}^M\}$: Reasonable personalized gate network models.

1: Server collects all the client's personalized experts and parameters to form a personalized pool $\mathcal{W}_{PE}, \mathcal{W}_{PP}$ and sends them to $M$ clients.
2: Server sends $W_{g,fe}, W_{g,hd}$ to $M$ clients.
3: **for** $j \in [M]$ in parallel **do**
4:        **local initialization**
5:    Client $j$ overwrites $W_{g,fe}, W_{g,hd}$ with the server parameters and freezes all of them.
6:    Client $j$ initializes the gated network $\theta_{PE}^j, \theta_{PP}^j$ for $\mathcal{W}_{PE}, \mathcal{W}_{PP}$ collected by the server.
7:    The MOE combination of $\{\theta_{PE}^j, \mathcal{W}_{PE}\}, \{\theta_{PP}^j, \mathcal{W}_{PP}\}$ replaces the local personalized experts and parameters.
8:        **local MOE learning**
9:    **for** $t = 0$ to $E_{moe}$ **do**
10:        Extract negative Helmholtz free energy $H^k(h^j, h^k)$ by Eq.12
11:        Removed $\gamma$-proportion part by Dropout($\gamma, \mathcal{W}_{PE}, \mathcal{W}_{PP}$)
12:        Client $j$ updates $\theta_{PE}^j, \theta_{PP}^j$ simultaneously:
13:        $\theta_{PE}^j \leftarrow \theta_{PE}^j - \eta_{moe} \nabla_{\theta_{PE}^j} G_{PE}^j$
14:        $\theta_{PP}^j \leftarrow \theta_{PP}^j - \eta_{moe} \nabla_{\theta_{PP}^j} G_{PP}^j$
15:    **end for**
16: **end for**
17: **return** $\{\theta_{PE}^1, \ldots, \theta_{PE}^M\}, \{\theta_{PP}^1, \ldots, \theta_{PP}^M\}$

---

## A.3  Dataset

We use public datasets to perform experiments and evaluate the performance of PM-MOE. The detailed descriptions and statistics of these datasets are as follows:

- **MNIST [26]** dataset is a widely used collection for handwritten digit recognition, compiled by the National Institute of Standards and Technology (NIST). It consists of 60,000 training images and 10,000 test images, each a 28x28 grayscale representation of digits from 0 to 9.

- **FMNIST [50]** is a dataset of fashion product images intended as a more challenging alternative to the traditional MNIST. It

contains 10 categories of clothing items, such as T-shirts, trousers, and sweaters, with 7,000 grayscale images per category. There are 60,000 training images and 10,000 test images, all at 28x28 pixels. Fashion MNIST presents a greater challenge in terms of image quality and diversity, featuring more background details and varying perspectives.

- **Cifar10 [23]** consists of 60,000 32x32 color images divided into 10 classes, with 6,000 images per class. Of these, 50,000 are used for training and 10,000 for testing. The dataset is split into five training batches and one test batch, each containing 10,000 images. The test batch includes 1,000 randomly chosen images from each class, while the training batches may have varying class distributions across batches.

- **Cifar100 [23]** dataset contains 60,000 32x32 color images, but it is divided into 100 classes, with 600 images per class. Each class has 500 images for training and 100 for testing. These 100 classes are grouped into 20 super-classes, with each image having both a "fine" label (its specific class) and a "coarse" label (its super-class).

- **TINY [7]** dataset is a subset of ImageNet, released by Stanford University. It comprises 200 classes, each with 500 training images, 50 validation images, and 50 test images. The images have been preprocessed and resized to 64x64 pixels and are commonly used in deep learning for image classification tasks.

- **AGNews [60]** dataset is an open dataset for text classification, containing 120,000 news headlines and descriptions from four categories: World, Sports, Business, and Technology. Each category includes 30,000 samples, with 120,000 samples in the training set and 7,600 in the test set.

## A.4 Baselines

In our experiments, the comparison baselines mainly include traditional federated learning methods (FedAvg, FedProx, SCAFFOLD, MOON, and FedGen), federated learning of personalized experts (FedGH, LG-FedAvg, FedBABU, FedCP, GPFL, FedPer, FedRep, FedRod), and federated learning of personalized parameters (DBE).

- **FedAvg [39]** is a pioneering algorithm in federated learning. Its core idea is to send the global model from the server to participating clients, where each client trains the model using their local data. The updated model parameters are then uploaded to the server, which computes the average of these parameters to update the global model. FedAvg can encounter performance bottlenecks when faced with highly imbalanced data or significant differences in client computing power.

- **FedProx [36]** aims to address the performance degradation of FedAvg when dealing with non-i.i.d. data. It introduces a regularization term during local training to penalize the deviation of model parameters from the global model, stabilizing the optimization process and preventing local models from straying too far from the global model.

- **SCAFFOLD [21]** tackles the issue of client drift by using control variates to reduce the variance between local updates and the global model. This ensures closer alignment between local models and the global objective, especially in non-i.i.d. data scenarios.

- **MOON [28]** is a federated learning algorithm based on contrastive learning. It aims to minimize the feature representation difference between the local and global models while maximizing the difference between successive local models. By contrasting the global and local model representations, MOON enhances the generalization ability of the global model in federated environments.

- **FedGen [62]** is a federated learning algorithm using knowledge distillation without data. It employs a lightweight generator on the server side to synthesize data, which is broadcasted to clients to assist their model training. This method not only optimizes the global model but also introduces inductive bias to local models, improving generalization in non-i.i.d. settings.

- **FedGH [53]** is a federated learning framework for heterogeneous models. It trains a shared Global Prediction Header (GPH) to integrate diverse model structures from different clients. The GPH is trained using feature representations extracted by clients' private feature extractors and learns global knowledge from various clients. The server then transmits the shared GPH to all clients, replacing their local prediction heads.

- **LG-FedAvg [32]** is a variant of FedAvg that trains both global and local models simultaneously. The global model acts as a classifier, while the local model is a feature extractor. During each iteration, both the classifier and feature extractor are updated concurrently without freezing any part of the model.

- **FedBABU [40]** updates only the body of the model during training, leaving the head randomly initialized and unchanged. This allows the global model to improve generalization during training, while the head is fine-tuned for personalization during evaluation, achieving efficient personalization with consistent performance improvements.

- **FedCP [58]** introduces conditional layers tailored to each client's data, which split the output of a shared extractor into personalized and global representations. The shared classifier handles global representations, while personalized classifiers manage personalized ones. Additionally, FedCP sets a regularization loss, ensuring that global feature representations remain as consistent as possible across rounds.

- **GPFL [56]** personalizes federated learning by incorporating personalized layers into the global model, capturing client-specific features. GPFL aims to adapt to each client's unique needs while maintaining privacy, making it suitable for scenarios with highly heterogeneous data distributions.

- **FedPer [4]** personalizes federated learning by keeping certain model layers (typically the final few) private to each client while sharing the remaining layers globally. This enables each client to fine-tune their local layers for personalized tasks while benefiting from the shared global model. Unlike FedBABU, the local classification head in FedPer is not frozen, and both the feature extractor and classification head are optimized during local training.

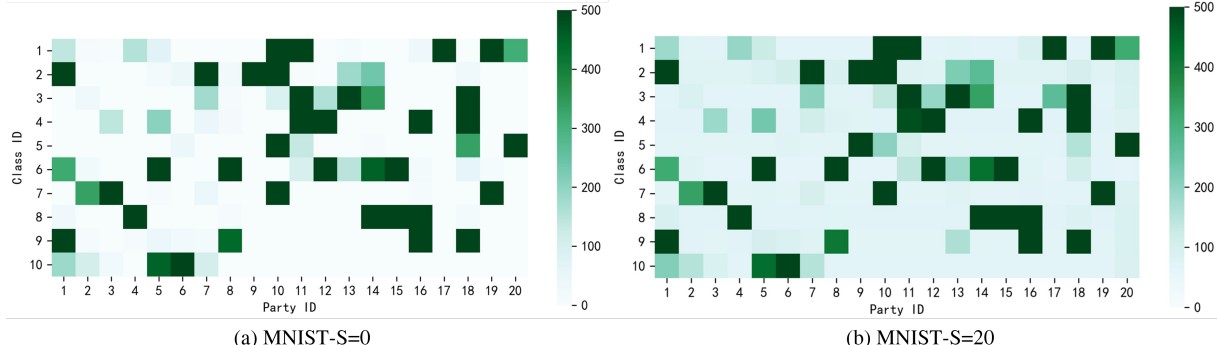

(a) MNIST-S=0                                              (b) MNIST-S=20

**Figure 9: Distribution of class data across all clients for the MNIST dataset.**

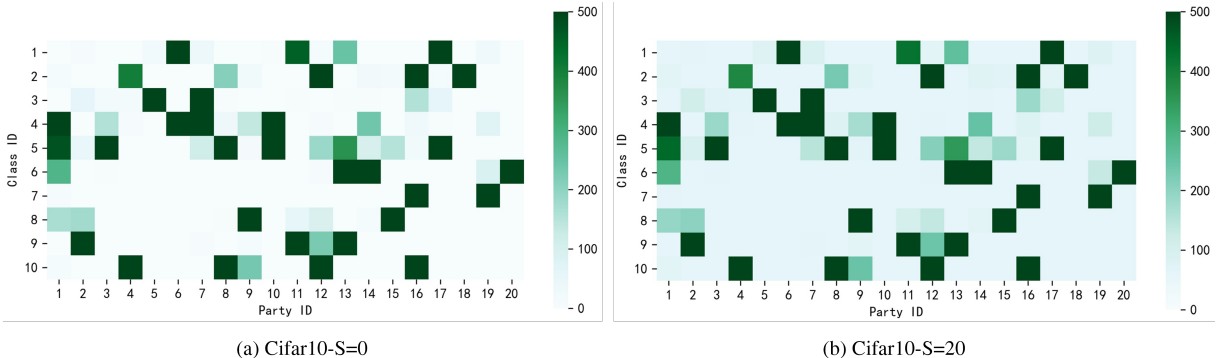

(a) Cifar10-S=0                                            (b) Cifar10-S=20

**Figure 10: Distribution of class data across all clients for the CIFAR-10 dataset.**

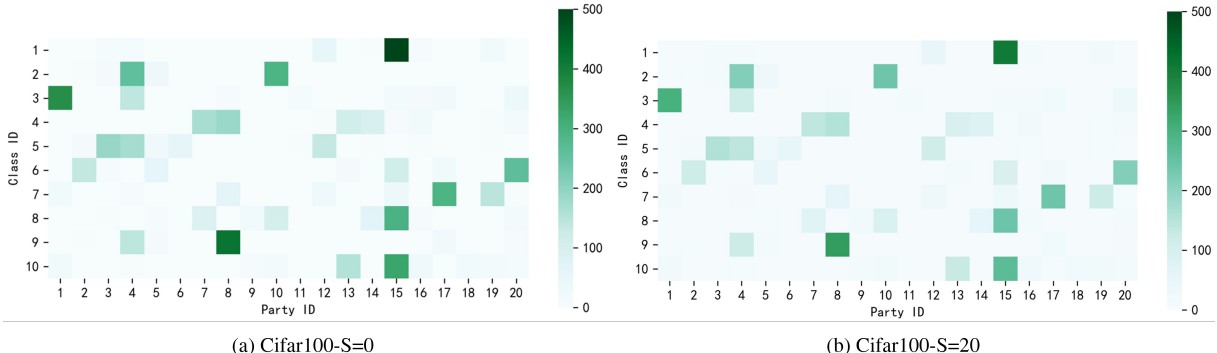

(a) Cifar100-S=0                                           (b) Cifar100-S=20

**Figure 11: Distribution of class data across all clients for the CIFAR-100 dataset (top 10 classes).**

- **FedRep [8]** first learns a shared representation through a matrix method, followed by alternating updates between clients and the server. FedRep demonstrates strong convergence in multilinear regression problems and significantly reduces sample complexity for new clients joining the system.

- **FedRod [6]** introduces a robust loss function that allows clients to train a universal predictor on non-identically distributed categories. It also includes a lightweight adaptive module (personalized classifier) that minimizes each client's empirical risk based on the shared universal predictor.

- **DBE [55]** is a method designed to tackle data heterogeneity in federated learning. It eliminates domain shifts in the representation space, optimizing the bidirectional knowledge transfer process between the server and clients. DBE sets a group of locally optimized private parameters to align and correct global model discrepancies.

## A.5 Evaluation Metrics

In personalized federated learning, the assessment of global accuracy can be formulated as the weighted sum of each client's accuracy

rate multiplied by its sample proportion. The formal expression is as follows:

$$A_{total} = \sum_{j=1}^{M} \frac{N^j}{N} \cdot A^j \qquad (31)$$

where $A_{total}$ denotes the weighted total accuracy. $M$ is the total number of clients. $A^j$ represents the accuracy of the $j-th$ client, and $N^j$ is the number of samples from the $j-th$ client. $N = \sum_{j=1}^{M} N^j$ is the total number of samples across all clients. $\frac{N^j}{N}$ signifies the proportion of samples from the $j-th$ client.

## A.6 Data Distribution Visualization

We demonstrate here the data distribution with S=0 and S=20 in our experiments. As illustrated in Figure 9, 10, 11, it can be observed that with S=20, the proportion of category samples across all clients is very low, with data evenly distributed among clients but in relatively small quantities. Consequently, classification performance for such data is generally poor. Each client particularly needs to leverage knowledge from other clients to enhance its performance, necessitating effective use of personalized models converged from other clients.

