# OpenReview forum: "PM-MOE: Mixture of Experts on Private Model Parameters for Personalized Federated Learning"
_ACM.org/TheWebConf/2025/Conference — WWW 2025 Oral_

### Official Review · Reviewer_nBfy · 2024-11-25

**Novelty:** 4
**Technical Quality:** 4

**Review:**

The author(s) present a method combining a mix of Experts, energy-based denoising, and Personalized Federated Learning to optimize the model's performance on each FL client. The proposed theories are demonstrated as a proof process in the paper. The method uses a Gate to generate a weight for fine tuning each client model. However, several issues need to be addressed for clarity and accuracy. Addressing these issues will significantly improve the clarity and rigor of the paper. Therefore, I vote for “major revision”
 * Strong points *
1. The author(s) propose a Mixture of personalized architecture to improve FL’s preference on each client by a weighting Gate.
2. An energy-based denoising module could provide each client with optimised-selected parameters.
3. The proposed theories are demonstrated as a proof process in the paper.

**Questions:**

* Weak points *
1. The main contribution, improvements of the proposed method and evaluation matrixes can be highlighted more straightforwardly in the abstract.
2. The author(s) named the proposed architecture PM-MOE; however, the PM is not clearly explained in the paper.
3.  To avoid gradient leaking, the clients exchange converged model parameters during training. However, other potential security risks are related to model tampering, model trust and validation, etc., which should also be discussed when the paper emphasizes cybersecurity and privacy. Alternatively, the paper should elaborate on the network security constraints specific to this experiment.
4. The paper provides Accuracy results to evaluate the PM-MOE’s performance. The Table 1 shows the proposed method has much better performance than other methods on some of certain datasets, and not many improvements on other datasets, which should be also discussed.
5. In Table 2 and 3, the ablation test demonstrates minor average improvements when other FL models are combined with the PM-MOE. Is the benefit worth the increased computational cost? Also, the robustness of the system combined with PM-MOE Should be discussed to proof it is an advanced architecture then others.

 * Typos and minors *
1 Type errors such as in line 265 “which which is a two-stage training framework.” should be corrected.

**Reviewer Confidence:**

3: The reviewer is confident but not certain that the evaluation is correct

**Scope:**

4: The work is relevant to the Web and to the track, and is of broad interest to the community

---

### Official Review · Reviewer_SQcs · 2024-11-29

**Novelty:** 5
**Technical Quality:** 3

**Review:**

This paper introduces the Mixture of Experts into the personalized federated algorithm while using gate network to select the corresponding experts. It was found that traditional MoE would to some extent affect the personalization performance, and they don't continue to practice the traditional Top-K experts ranking method. The Energy-based Personalized Modules Denoising was proposed to solve this problem.

**Hightlights**:
- They innovatively take the energy-denoise to do the Ranking elimination system for MoE. This measure could mitigate the Non-IID in personalization.
- Experiments result demonstrate the effectiveness of the algorithm.  In addition, they did a lot of scalable experiments to prove the effectiveness.
- They provide usable, verifiable algorithm code.
- Their proof focus on Bound for MPE prediction and the proof process is correct.
- Inserted pictures are fancy.

**Drawbacks**:
- Part of pictures don't have any descriptions, and isn't easy to capture all the info in quick.
- The description in this article is not good. Many descriptions are absent or lack evidence.
- Definitions and notations for some entities required to be refined.

**Others**:
There are quite a few typos, although it does not affect the overall reading.

**Questions:**

- pre-training step, why do you train for converged global and personalized meanwhile? whether global one is uploaded to the server for aggregation? And what is the difference of the training process for global and personalization? Is it DBE?

> we train model and obtain its converged global and personalized modules for each client, thereby constructing a personalized prompt pool

- $x_{rep}^{j} = f_g{}(w_{g,fe}) + ...$ what is the $fe$? and what about the following $hd$?
- 3.2 Mixture of Personalized Modules

> *Assume that a personalized federated learning algorithm involves personalized parameters, these parameters do not project data into vectors of other dimensions. We define this type as local personalized parameters (PP).*

What does this sentence mean? whether your PP train sth or not?
- what does $x_{moe}^{j}$ mean? it's the parameter vector or input data?
- Where we got this initializing model? Like (PE), Is it from the pre-trained step?
- Put them all in a pool, the models in this pool is from pre-trained right?
- You leverage the input x and $\theta_pp$ to go through gating network, get the weight of each $\theta_pp$, and then do the weight aggregation and get the client of $pp_{moe}$. Please check whether this step is correct.
- I really don't know the connection between PP and PE, if PP doesn't do any inference, why do we need to gate it?
- In your experiments, whether PP does the inference or both PE and PP do? I believe that all the definitions truly require rewriting along with additional clarification.
- Why don't you compare with the FedMoE?
- Your performance is average client performance or the global model performance?

**Reviewer Confidence:**

4: The reviewer is certain that the evaluation is correct and very familiar with the relevant literature

**Scope:**

4: The work is relevant to the Web and to the track, and is of broad interest to the community

---

### Official Review · Reviewer_BZk8 · 2024-11-30

**Novelty:** 6
**Technical Quality:** 6

**Review:**

### Review of the Paper: "PM-MOE: Mixture of Experts on Private Model Parameters for Personalized Federated Learning"

#### **1. Problem Addressed**

The paper tackles a significant problem in federated learning (FL): how to improve the generalization of models in heterogeneous data settings while maintaining privacy. It addresses the shortcomings of traditional personalized federated learning (PFL) methods, which often fail to leverage domain-specific personalized parameters across clients effectively. This is a timely and important problem given the growing interest in privacy-preserving collaborative machine learning.

#### **2. Novelty and Contributions**

The proposed PM-MOE framework is innovative, introducing two key contributions:

- **Mixture of Personalized Modules (MPM):** A dynamic weighting mechanism via gating networks that allows clients to selectively incorporate knowledge from other clients' personalized modules.
- **Energy-Based Denoising Method (EDM):** A novel denoising approach that filters out irrelevant or noisy parameters to enhance performance while maintaining privacy.

These contributions are highly relevant and address critical gaps in existing PFL methods.

#### **3. Strengths**

1. **Innovative Approach:** The use of MoE in the PFL context, combined with energy-based denoising, is a novel and impactful idea.
2. **Comprehensive Analysis:** Theoretical guarantees and empirical results both support the claims, providing a well-rounded argument for the framework's effectiveness.
3. **Privacy-Preserving Design:** The method ensures that private data remains local, and the use of converged parameters avoids gradient leakage.
4. **Scalability:** The framework is evaluated in various data heterogeneity settings, indicating its adaptability to real-world scenarios.

#### **4. Weaknesses**

1. **Scalability to Large-Scale Systems:**
   - While the framework performs well on benchmark datasets, it is unclear how it scales to larger datasets or systems with hundreds of clients.
   - The computational overhead of the gating network and denoising method is not addressed in detail.
2. **Denoising Assumptions:**
   - The energy-based denoising relies on similarity measures that may not generalize well across drastically different domains.
   - Its impact in extremely noisy or adversarial settings is unexplored.
3. **Implementation Details:**
   - Although the methodology is clear, more details on hyperparameter selection and tuning (e.g., gating network initialization) would enhance reproducibility.
4. **Broader Applicability:** The paper focuses on model-splitting-based PFL methods. It would be interesting to see if the approach generalizes to other FL paradigms (e.g., hierarchical FL or cross-device FL).

**Overall:** The paper is a valuable addition to the field and should be considered for publication, pending minor clarifications and additional analysis.

**Questions:**

#### **Weaknesses**

1. **Scalability to Large-Scale Systems:**
   - While the framework performs well on benchmark datasets, it is unclear how it scales to larger datasets or systems with hundreds of clients.
   - The computational overhead of the gating network and denoising method is not addressed in detail.
2. **Denoising Assumptions:**
   - The energy-based denoising relies on similarity measures that may not generalize well across drastically different domains.
   - Its impact in extremely noisy or adversarial settings is unexplored.
3. **Implementation Details:**
   - Although the methodology is clear, more details on hyperparameter selection and tuning (e.g., gating network initialization) would enhance reproducibility.
4. **Broader Applicability:** The paper focuses on model-splitting-based PFL methods. It would be interesting to see if the approach generalizes to other FL paradigms (e.g., hierarchical FL or cross-device FL).

**Reviewer Confidence:**

3: The reviewer is confident but not certain that the evaluation is correct

**Scope:**

1: The work is irrelevant to the Web

---

### Official Review · Reviewer_cdmC · 2024-12-03

**Novelty:** 6
**Technical Quality:** 6

**Review:**

The paper proposes a novel framework, PM-MOE, which effectively integrates a Mixture of Experts (MoE) structure with energy-based denoising for personalized federated learning (PFL). This dual mechanism addresses significant challenges in heterogeneous data distributions by leveraging personalized models across clients.
The framework ensures privacy by localizing sensitive personalized parameters and effectively preventing gradient leakage during training. Additionally, the framework introduces minimal computational overhead, making it a scalable solution for PFL.
The inclusion of theoretical proofs, such as convergence guarantees and the derivation of lower bounds for performance improvements, strengthens the claims of the paper.
The flexibility of PM-MOE to integrate with existing PFL algorithms without altering their architectures is a significant advantage, showcasing the framework's adaptability to real-world applications.

**Questions:**

1. Sharing and selecting personalized parameters from multiple clients could introduce communication overhead. A quantitative evaluation of this aspect would enhance the framework's feasibility.

2. Although the framework introduces minimal training overhead, a detailed analysis of its scalability to large-scale datasets or real-world deployment scenarios is missing.

**Reviewer Confidence:**

4: The reviewer is certain that the evaluation is correct and very familiar with the relevant literature

**Scope:**

4: The work is relevant to the Web and to the track, and is of broad interest to the community

---

### Official Review · Reviewer_FsrH · 2024-12-05

**Novelty:** 6
**Technical Quality:** 5

**Review:**

The submitted paper proposes the PM-MoE architecture, which integrates a mixture of personalized modules and energy-based personalized module denoising, enabling each client to select beneficial personalized parameters from other clients. Extensive experiments on six widely adopted datasets and two heterogeneity settings validate the effectiveness of our approach. The problem mentioned in this paper is an interesting work, however, there are some suggestions for improving the paper.
1)In the citations of references, it is recommended not to cite too many journals at once, for example: "federated learning (FL) [1, 5, 15, 35, 39, 52], knowledge distillation [43–45, 49, 51]."
2)For the second point of contribution, it is recommended that the authors carefully refine it to emphasize the innovation of the article.
3)The loss function of PM-MOE should be further elaborated to help readers better understand it.
4)In the ablation study section, what does each module represent? Additionally, the presentation of experimental results should not only focus on the performance improvement but also analyze the reasons behind it. Furthermore, with a performance increase of only 0.2, did the authors consider that the corresponding time cost has also increased significantly?
5)The pseudocode for the algorithm and its time complexity analysis should be provided.
6)The limitations of the work presented in this paper, as well as the connection between future work and the work discussed herein, should be elaborated.

**Questions:**

Furthermore, with a performance increase of only 0.2, did the authors consider that the corresponding time cost has also increased significantly?

**Reviewer Confidence:**

3: The reviewer is confident but not certain that the evaluation is correct

**Scope:**

3: The work is somewhat relevant to the Web and to the track, and is of narrow interest to a sub-community